# Carbapenem Combinations for Infections Caused by Carbapenemase-Producing *Pseudomonas aeruginosa*: Experimental In Vitro and In Vivo Analysis

**DOI:** 10.3390/antibiotics11091212

**Published:** 2022-09-07

**Authors:** Soraya Herrera-Espejo, Ester Del Barrio-Tofiño, Tania Cebrero-Cangueiro, Carla López-Causapé, Rocío Álvarez-Marín, José Miguel Cisneros, Jerónimo Pachón, Antonio Oliver, María Eugenia Pachón-Ibáñez

**Affiliations:** 1Unit of Infectious Diseases, Microbiology, and Preventive Medicine, Virgen del Rocío University Hospital, 41013 Seville, Spain; 2Institute of Biomedicine of Seville (IbiS), Virgen del Rocío University Hospital/CSIC/University of Seville, 41013 Seville, Spain; 3Microbiology Service, Son Espases University Hospital, Instituto de Investigación Sanitaria Illes Balears (IdISBa), 07010 Palma de Mallorca, Spain; 4CIBER de Enfermedades Infecciosas, Instituto de Salud Carlos III, 28220 Madrid, Spain; 5Department of Medicine, University of Seville, 41004 Seville, Spain

**Keywords:** carbapenemase-producing *Pseudomonas aeruginosa*, doripenem, efficacy studies, imipenem, meropenem, murine sepsis model

## Abstract

In the context of difficult-to-treat carbapenem-resistant *Pseudomonas aeruginosa* infections, we evaluated imipenem, meropenem, and doripenem combinations against eleven carbapenemase-producing *P. aeruginosa* isolates. According to the widespread global distribution of high-risk clones and carbapenemases, four representative isolates were selected: ST175 (OXA-2/VIM-20), ST175 (VIM-2), ST235 (GES-5), and ST111 (IMP-33), for efficacy studies using a sepsis murine model. Minimum inhibitory concentration (mg/L) ranges were 64–256 for imipenem and 16–128 for meropenem and doripenem. In vitro, imipenem plus meropenem was synergistic against 72% of isolates and doripenem plus meropenem or imipenem against 55% and 45%, respectively. All combinations were synergistic against the ST175, ST235, and ST155 clones. In vivo, meropenem diminished the spleen and blood bacterial concentrations of four and three isolates, respectively, with better efficacy than imipenem or doripenem. The combinations did not show efficacy compared with the more active monotherapies, except for imipenem plus meropenem, which reduced the ST235 bacterial spleen concentration. Mortality decreased with imipenem plus meropenem or doripenem for the ST175 isolate. Results suggest that carbapenem combinations are not an alternative for severe infections by carbapenemase-producing *P. aeruginosa*. Meropenem monotherapy showed in vivo efficacy despite its high MIC, probably because its dosage allowed a sufficient antimicrobial exposure at the infection sites.

## 1. Introduction

*Pseudomonas aeruginosa* is a common cause of community-acquired infections in patients with chronic underlying diseases and hospital-acquired infections, such as pneumonia, urinary tract infections, and bloodstream infections (BSIs) [1]. Recently, it has been described as a pathogen that co-infects patients with COVID-19 [2,3]. The 2016 report by the ECDC on infections acquired in the ICU in 18 European countries showed that, in 2014, *P. aeruginosa* was the most common cause of ventilator-associated pneumonia and the fifth most prevalent in ICU-acquired BSI [4]. *P. aeruginosa* infections are associated with elevated disease burden and mortality rates in the absence of optimal treatment [5]. A prospective study showed that patients with *P. aeruginosa* BSI present higher mortality rates than those with Enterobacteriaceae or other non-lactose fermenting Gram-negative bacilli (GNB) infections [6]. 

Carbapenem-resistant *P. aeruginosa* is a significant health concern listed by the WHO as a “priority pathogen”; therefore, it is urgent and necessary to search for new treatments [7]. The European Antimicrobial Resistance Surveillance Network (EARS-Net) reported that 16.9%, 18.9%, 14.5%, and 11.5% of analyzed *P. aeruginosa* isolates were non-susceptible to ceftazidime, fluoroquinolones, and imipenem/meropenem, and demonstrated combined resistance to three or more antimicrobial groups (including piperacillin/tazobactam, ceftazidime, carbapenems, fluoroquinolones, and aminoglycosides), respectively (2019 data from 30 European Union countries) [8]. Moreover, due to the increased use of carbapenems to treat multidrug-resistant (MDR) *P. aeruginosa* infections, there is also a growth in carbapenemase production among carbapenem-resistant *P. aeruginosa* isolates, including the Ambler class A KPC- and GES-type beta-lactamases and the Ambler class B or metallo-beta-lactamases (MBLs), mainly VIM (Verona integrin-encoded MβL), IMP (imipenemase), and NDM (New Delhi MβL) types [9]. In Spain, according to EARS-Net, resistance to carbapenems in invasive isolates of *P. aeruginosa* has increased up to 21.8% in 2019 [8]. 

A recent study in India showed that most hospital-acquired *P. aeruginosa* isolates were MDR [10]; moreover, carbapenem-resistant *P. aeruginosa* (CRPA) frequently produced carbapenemase, with 62.7% of isolates having at least one carbapenemase gene [11]. 

The production of MBLs, especially VIM and IMP, has increased in *P. aeruginosa* isolates [12,13]. Dissemination of high-risk clones, such as ST175 and ST244, producing class B carbapenemases (VIM and IMP) caused outbreaks [14,15,16]. Carbapenemase-producing *P. aeruginosa* is correlated with higher morbidity and mortality. Health services urgently need to focus on infection prevention and adequate infection management by developing new therapies for these infections [17]. 

Recent studies found that the association of two carbapenems is efficacious both in vitro and in sepsis models in mice against carbapenemase-producing *Acinetobacter baumannii* [18,19], multidrug-resistant Enterobacteriaceae [20], KPC-producing *Klebsiella pneumoniae* [21], and KPC-producing *Escherichia coli* [22]. In the search for new alternatives for difficult-to-treat carbapenem-resistant *P. aeruginosa* infections, we aimed to evaluate whether a dual carbapenem treatment could be useful in infections by carbapenemase-producing *P. aeruginosa*.

## 2. Results

### 2.1. In Vitro Results

#### 2.1.1. Isolates’ Carbapenemase Production, Molecular Typing, Antimicrobial Susceptibility Testing, and FICi of Dual Carbapenem Combinations 

The sequence types, carbapenemases produced, and MICs of each antibiotic for the eleven isolates are shown in Table 1. All isolates were resistant to the three carbapenems. The checkerboard assays show synergy of meropenem plus doripenem against three strains (CAT05-004 (VIM-1), ST253; MAD05-041 (VIM-2), ST111; and MUR01-018 (IMP-33), ST111); meropenem plus imipenem against MAD02-007 (VIM-2), ST175; and imipenem plus doripenem against CAT05-004 (VIM-1), ST253. 

#### 2.1.2. Time-Kill Assays

With the assays carried out at MIC concentrations, doripenem was bactericidal against 55% (6/11), imipenem against 36% (4/11), and meropenem against 9% (1/11) of the isolates. The three combinations of carbapenems increased activity compared to when they were tested alone. Bactericidal activity occurred against 9 out of the 11 isolates—the exceptions were ARA01-045 (VIM-2), ST973 and MAD04-041 (IMP-8), ST155. The combination of imipenem plus meropenem was synergistic against 72% (8/11) of isolates, followed by doripenem plus meropenem or imipenem against 55% (6/11) and 45% (5/11), respectively. All combinations were bactericidal and synergistic against the high-risk ST175 (CVA03-019 (OXA2/VIM-20), MAD02-005 (GES-5), and MAD02-007 (VIM-2)) and the ST235 (MAD02-021 (GES-5)) isolates. Finally, for the *P. aeruginosa* ST973 (ARA01-045 (VIM-2)) isolate, no carbapenem alone or in combination showed bactericidal activity or synergy (Figure 1a,b and Appendix A).

The time-kill assays at *C*_max_ concentrations were performed against the four isolates selected for the in vivo experiments: CVA03-019 (OXA-2/VIM-20), ST175; MAD02-007 (VIM-2), ST175; MAD02-021 (GES-5), ST235; and MUR01-018 (IMP-33), ST111. Imipenem alone was not bactericidal against any of the four isolates, as expected due to the lower value of *C*_max_ with respect to its MIC. On the contrary, meropenem and doripenem, with *C*_max_ higher than their respective MICs, were bactericidal against the four isolates. The combinations of the carbapenems at *C*_max_ were not synergistic (Figure 2a,b).

### 2.2. In Vivo Results

#### 2.2.1. Peritoneal Sepsis Model 

The MLDs obtained for the peritoneal sepsis model in mice were: 8.28, 9.55, 7.11, and 8.66 log_10_ CFU/mL for CVA03-019 (OXA-2/VIM-20), MAD02-007 (VIM-2), MAD02-021 (GES-5), and MUR01-018 (IMP-33), respectively. Bacterial loads in the spleen and blood and the frequency of bacteremia at initiation of treatment and at 24 h are detailed in Table 2.

#### 2.2.2. Efficacy Studies

The in vivo efficacies of imipenem, meropenem, and doripenem in monotherapy and combination against the four selected isolates are detailed in Table 3. Meropenem monotherapy diminished the spleen bacterial concentrations in control mice for all isolates between −2.0 and −4.9 log_10_ CFU/g; similarly, in blood, it diminished bacterial concentrations in three isolates between −3.7 and −5.9 log_10_ CFU/mL, the exception being the MUR01-018 (IMP-33) isolate. Meropenem monotherapy was better than imipenem in decreasing the spleen and blood bacterial concentrations in the infection with three and two isolates, respectively. Furthermore, meropenem monotherapy was better than doripenem against two and one isolates, respectively. 

Imipenem monotherapy diminished the spleen and blood bacterial concentrations only for the MUR01-018 (IMP-33) isolate (−2.9 log_10_ CFU/g) and for the MAD02-021 (GES-5) and MUR01-018 (IMP-33) (−3.3 and −2.1 log_10_ CFU/mL) isolates, respectively. Doripenem monotherapy diminished only the spleen bacterial concentrations for the MAD02-021 (GES-5) and MUR01-018 (IMP-33) (−3.8 and −1.7 log_10_ CFU/g) isolates. Regarding the mortality rate, the three carbapenems in monotherapy increased the survival for the MAD02-021 (GES-5).

The three combinations of two carbapenems were efficacious in reducing the bacterial concentrations in spleen and blood, compared with their respective controls, against the four *P. aeruginosa* clinical isolates producing OXA-2/VIM-20, VIM-2, GES-5, or IMP-33. The exceptions were the bacterial concentration in blood for the IMP-33 producer with imipenem plus meropenem, the bacterial spleen concentration for the VIM-2 producer with imipenem plus doripenem, and the bacterial spleen and blood concentrations for the VIM-2 and IMP-33 producers with meropenem plus doripenem. However, the combinations of two carbapenems did not show efficacy in general compared with the more active monotherapies. Only the combination of imipenem plus meropenem reduced the spleen bacterial concentration, compared with the most efficacious monotherapy, for the GES-5 producer isolate (−2.9 log_10_ CFU/g). Regarding mortality, it decreased with imipenem plus meropenem or doripenem for the OXA-2/VIM-20 producer isolate (83% vs. 0%) compared with the monotherapies.

## 3. Discussion

To the best of our knowledge, dual carbapenem combinations have never been tested in vivo against carbapenemase-producing *P. aeruginosa*. In this study, we found that the combination of carbapenems was not more efficacious than the most active carbapenem monotherapy in a murine sepsis model infected with *P. aeruginosa* clinical isolates producing OXA-2/VIM-20, VIM-2, GES-5, or IMP-33. Unexpectedly, meropenem monotherapy showed in vivo efficacy despite its high MIC against the tested isolates in reducing bacterial concentrations in the spleen and blood and was better than imipenem or doripenem. Meropenem in monotherapy was also able to reduce mortality compared to control mice against the GES-5 producer. These results with meropenem monotherapy are probably associated with its serum concentration exceeding the MIC for more than 50% of the time between doses (with the dosage used), which is equivalent to a dosage in humans of 2 g every eight hours [24]. A pharmacokinetics/pharmacodynamics (PK/PD) model using a Monte Carlo simulation [25] supports this hypothesis by showing that an optimized administration of meropenem at a dose of 2 g/8 h achieved a PD target ≥90% for *P. aeruginosa* isolates with an MIC of up to 128 mg/L, similar to the isolates used in our in vivo experiments, with a cumulative fraction of response ≥90%. These results with meropenem reinforce the category of “susceptible increased exposure” released by the EUCAST in 2021 [26]. The efficacy of this dosage was also observed in a recent study in a murine sepsis model by carbapenemase-producing *A. baumannii* [19]. 

Imipenem monotherapy reduced the bacterial concentration in the spleen and blood in the in vivo experiments with the IMP-33 producer isolate and in blood with the GES-5 producer. The efficacy of imipenem monotherapy may not be associated with the pharmacodynamics of the dosage used, based on the reported pharmacokinetics (PK) data [27]. In the case of the GES-5 producer isolate, the in vivo efficacy mimics the time-kill in vitro results, in which imipenem at concentrations equivalent to its MIC was bactericidal at 4 and 8 h. However, its in vivo efficacy against the IMP-33 producer may not be related to the in vitro results. In this case, the results with imipenem and doripenem may be related to the low virulence of the IMP-33 producer isolate in terms of lesser bacterial concentrations in the spleen and blood and its lower mortality compared with the other three isolates tested in vivo. The efficacy of imipenem was also observed in a murine model against OXA-58 and OXA-23 *A. baumannii* producer strains [19]. Similarly, doripenem monotherapy showed efficacy in reducing the spleen bacterial concentrations against the GES-5 and the IMP-33 producer isolates, respectively, thereby mimicking the bactericidal activity observed in vitro in the time-kill studies, besides the low virulence of the IMP-33 isolate. It must be mentioned that, currently, doripenem is available in very few countries.

The three combinations of two carbapenems were generally efficacious in reducing the bacterial concentrations in the spleen and blood compared with their respective controls against the four *P. aeruginosa* clinical isolates producing OXA-2/VIM-20, VIM-2, GES-5, or IMP-33. However, the combinations of two carbapenems were not better than the most efficacious monotherapy in reducing the spleen bacterial concentration for the GES-5 producer isolate, apart from imipenem plus meropenem. These in vivo results were not in agreement with the in vitro studies, which showed synergistic activity of the three combinations against the OXA-2/VIM-20, VIM-2, and GES-5 producer isolates, as well as the imipenem plus meropenem combination against the IMP-33 producer isolate. Besides the impact of the innate immune response and the pharmacodynamics of the in vivo experiments, the in vitro studies were performed with MIC concentrations higher than those achieved in vivo, considering the carbapenem resistance of the isolates. The good results achieved with the carbapenem combinations compared with the untreated controls were also reported against OXA-58 and OXA-23 *A. baumannii* producers [18,19]. 

The lack of proven valid therapeutic options and the rapid development of antimicrobial resistance for infections caused by carbapenemase-producing *P. aeruginosa* isolates justifies the use of combination therapy for infections caused by carbapenemase-producing pathogens [28]. Thus, in vitro synergism between meropenem and ertapenem has been reported against carbapenem-resistant *K. pneumoniae* [29]. Further combinations include meropenem plus doripenem or ertapenem against OXA-181 and VIM *E. coli* producers, NDM-1 *K. pneumoniae* producer isolates, and the combination of imipenem plus meropenem against the NDM-1 *K. pneumoniae* producer isolate [30]. The combination of two synergistic carbapenems, alone or combined with other families of antibiotics, has been suggested as a treatment for carbapenem-resistant, carbapenemase-producing *K. pneumoniae* [31] and carbapenemase-producing *A. baumannii* [19]. In addition to the different in vitro studies showing activity with dual carbapenem combinations [18,29,32], there have been positive clinical outcomes reported with meropenem plus ertapenem in BSI by KPC-producing *E. coli* [22] and by carbapenemase-producing *K. pneumoniae* [20], as well as reports that ertapenem plus meropenem or doripenem was efficacious in BSI and urinary tract infection by KPC-producing *K. pneumoniae* [21]. 

Although there are more experimental and clinical studies about the use of dual carbapenem therapies against infections by MDR-GNB published every day, the safety of these approaches has not been widely addressed. Nevertheless, there are some studies that have evaluated both the efficacy and safety of carbapenem combination treatments, suggesting that these treatments may be effective and safe to treat carbapenemase-producing *K. pneumoniae* [33] or carbapenem-resistant Enterobacteriaceae [34].

Although the exact mechanism of action is not fully understood against KPC-producing *K. pneumoniae*, some have postulated that the mechanisms for dual carbapenems against carbapenemase-producing organisms could be related to the binding of one of the carbapenems to the active site of the enzyme, using its action in a concentration-independent way, while the other carbapenem binds to the bacterial target [21,29]. Others have suggested that the benefit of the ertapenem plus doripenem combination could be related to the preferential affinity of the enzyme for ertapenem. In this way, the enzyme is consumed during the hydrolysis of ertapenem, leaving higher concentrations of doripenem acting against the pathogen [35]. 

Although our results suggest that these combinations are not an alternative for severe infections by carbapenemase-producing strains of *P. aeruginosa*, as in any animal model study, a limitation is the general caution to translate the preclinical studies to the clinical setting, although the antibiotics dosages have been chosen according to the pharmacodynamic targets in human beings. Moreover, the 3R rules [36] (Hubrecht and Carter, 2019) prevent us from increasing the numbers of animals to test the same hypothesis in male mice. As strengths of the study, the chosen isolates from the GEMARA/REIPI collection [23], both for the in vitro and in vivo studies, are representative of high-risk clones and widespread acquired carbapenemases [15,37,38,39]. 

## 4. Conclusions

The results suggest that dual carbapenem combinations are not a therapeutic alternative for severe infections by carbapenemase-producing strains of *P. aeruginosa*. Moreover, the in vitro activity of carbapenem combinations against class A and B carbapenemase producers did not match the in vivo results. Thus, the highest efficacy was observed against the ST 235, GES-5 producer isolate, despite its being the most efficient binding and catalyzing carbapenem from class A carbapenemases [40]. Unexpectedly, meropenem monotherapy showed in vivo efficacy despite its high MIC against the tested isolates, probably because its dosage allowed a sufficient antimicrobial exposure at the infection sites.

## 5. Materials and Methods

### 5.1. In Vitro Studies

#### 5.1.1. Bacterial Isolates’ Characterization and Molecular Typing

The GEMARA/REIPI study group collected healthcare-associated non-duplicated *P. aeruginosa* clinical isolates from 51 participating Spanish hospitals in 2017 [23], identified by a Microflex LT-MALDI Biotyper mass spectrometer (Bruker Daltonics GmbH, Bremen, Germany). The presence of carbapenemase genes and genes coding for other beta-lactamases was confirmed by PCR and sequencing; moreover, representative XDR isolates and PFGE clonal types were fully sequenced (Miseq, Illumina, La Jolla, CA, USA) [23]. Eleven of such isolates producing diverse carbapenemases and belonging to different clonal types were selected for the in vitro studies: ARA01-015 (VIM-2), ST235; ARA01-045 (VIM-2), ST973; CAT05-004 (VIM-1), ST253; CLE02-006 (IMP-1), ST664; CVA03-019 (OXA-2/VIM-20), ST175; MAD02-005 (GES-5), ST175; MAD02-007 (VIM-2), ST175; MAD02-021 (GES-5), ST235; MAD04-041 (IMP-8), ST155; MAD05-041 (VIM-2), ST111, and MUR01-018 (IMP-33), ST111. According to the widespread global distribution of *P. aeruginosa* high-risk clones and their association with acquired carbapenemases [15,16], the following four strains were selected for preclinical studies in mice, CVA03-019 (OXA-2/VIM-20), ST175; MAD02-007 (VIM-2), ST175; MAD02-021 (GES-5), ST235; and MUR01-018 (IMP-33), ST111.

#### 5.1.2. Antimicrobials

For the in vitro experiments, antibiotic drugs were used as standard laboratory powders: imipenem and doripenem from Alsachim (Alsachim, Illkirch Graffenstaden, France) and meropenem from Sigma (Sigma-Aldrich, Madrid, Spain). 

For the in vivo studies, clinical formulations were used for imipenem (Fresenius Kabi, Barcelona, Spain) and meropenem (Ranbaxy, Barcelona, Spain). In the case of doripenem, as it is not used in the clinic setting in Europe, it was used as a standard powder from Alsachim (Alsachim, Illkirch Graffenstaden, France).

#### 5.1.3. Antimicrobial Susceptibility Testing 

MICs of the three carbapenems were determined by the broth microdilution method with geometric two-fold serial dilutions of the antimicrobial agents (ranging from 2 to 1024 mg/L), as recommended by the EUCAST [26]. Tests were conducted using Cation-adjusted Müller Hinton Broth II (CAMHBII, 90922 Merck Life Science S.L., Madrid, Spain) and a final inoculum of 5 × 10^5^ CFU/mL. Quantification of the initial inoculum and bacterial growth was obtained by subculture on blood agar plates (Becton Dickinson) incubated for 18–24 h at 37 °C in air. *P. aeruginosa* ATCC 27853 was used as a quality control strain. MICs were evaluated after 24 h of incubation at 37 °C. The MIC was defined as the lowest concentration of antibiotic at which no growth was visible. Results were interpreted according to the EUCAST breakpoints [26]. 

#### 5.1.4. Synergy Studies

##### Checkerboard Assays

The antibiotic concentrations studied were from 8 to 512 mg/L for imipenem and from 4 to 256 mg/L for meropenem and doripenem, respectively. A two-fold dilution of each antimicrobial agent alone or with different combinations was performed in CAMHBII in 96-wells microplates. A freshly prepared inoculum of each isolate was added to obtain a concentration of 5 × 10^5^ CFU/mL in each well. Plates were incubated at 37 °C for 18–24 h, and the MIC was read. Growth and sterility controls were also included in each plate. The reference strain *P. aeruginosa* ATCC27853 was used as a control. The assay was performed in triplicates for each antibiotic.

Fractional inhibitory concentration indexes (FICi) were calculated according to the following formula, FICi = FIC of antibiotic A + FIC of antibiotic B where FIC of A or B = MIC of A or B in combination, divided by the MIC of A or B alone. Interpretation of the results was based on the following FICi values: synergy ≤ 0.5, no interaction > 0.5 to 4, and antagonism > 4 [18]. 

##### Time-Kill Assays

Imipenem, meropenem, and doripenem were tested at MIC values alone and in combination for the eleven isolates. Moreover, for the four isolates selected for the peritoneal sepsis model, time-kill assays were also performed at the peak concentrations achievable in serum. The studies were carried out in log-phase with a starting inoculum of 5 × 10^5^ CFU/mL with the antibiotics alone or in combination. Tubes were incubated at 37 °C with shaking, and samples were taken at 0, 2, 4, 8, and 24 h, serially diluted, and seeded in 5% sheep blood plates [41]. Bactericidal activity was defined as a ≥3 log_10_ CFU/mL decrease from the initial inoculum. Synergistic activity was defined as a decrease of ≥2 log_10_ CFU/mL for the antimicrobial combination compared with the most active single agent [42]. 

### 5.2. In Vivo Studies

#### 5.2.1. Animals

A total of 165 immunocompetent C57BL/6J female mice weighing 20 g (7–9 weeks old) were used (Production and Experimentation Animal Centre, University of Seville, Seville, Spain). Mice were housed in a ventilated cage system under specific pathogen-free conditions, with water and food ad libitum. The study was carried out following the recommendations in the Guide for the Care and Use of Laboratory Animals [43]. All efforts were made to minimize suffering. This study was approved by the Committee on the Ethics of Animal Experiments of the University Hospital Virgen del Rocío and the Ministry of Agricultura, Pesca, and Desarrollo Rural (08/03/2019/028), Spain. Mice were sacrificed using an intraperitoneal (ip) lethal dose of sodium thiopental (B. Braun Medical S.A., Barcelona, Spain). 

#### 5.2.2. Peritoneal Sepsis Model 

An experimental peritoneal sepsis model was used [27] with the four selected *P. aeruginosa* isolates based on their ST and carbapenemase production: CVA03-019 (OXA-2/VIM-20), ST175; MAD02-007 (VIM-2), ST175; MAD02-021 (GES-5) ST235; and MUR01-018 (IMP-33), ST111. The minimum lethal doses (MLDs, concentration of inoculum killing 100% of the animals) were determined by the Reed and Munch method [44] for the four strains. Briefly, for each strain, groups of six unanesthetized mice were ip inoculated with 0.5 mL of decreasing bacterial concentrations. The inoculum ranges used were 9.28 to 7.49 log_10_ CFU/mL for *P. aeruginosa* CVA03-019; 9.55 to 7.41 log_10_ CFU/mL for *P. aeruginosa* MAD02-007; 9.02 to 7.11 log_10_ CFU/mL for *P. aeruginosa* MAD02-021; and 8.66 to 7.56 log_10_ CFU/mL for *P. aeruginosa* MUR01-018. After inoculation, animals were observed and monitored for 7 days. 

#### 5.2.3. Efficacy Studies 

After the peritoneal sepsis model characterization for each isolate, we evaluated the efficacy of dual carbapenem treatment. Concisely, groups of mice (n = 5–6) were ip inoculated with 500 µL of the MLD (log_10_ CFU/mL) of each isolate. Treatments were initiated two hours after inoculation and lasted 24 h. Infected mice were randomly assigned to the followings groups: (i) control (untreated); (ii) imipenem 30 mg/kg/q4h administered intramuscularly (im); (iii) meropenem 300 mg/kg/q2h/ip; (iv) doripenem 150 mg/kg/q12h/ip; (v) imipenem plus meropenem; (vi) imipenem plus doripenem; and (vii) meropenem plus doripenem. Antimicrobials were used in combination with the same dosage schedule as in monotherapy, receiving the first dose of both carbapenems two hours after inoculation and the remaining doses with the specified time intervals. The antimicrobial dosages were based on the PK/PD data and their proven efficacy, alone and in combination, in experimental models of infected mice [19,24,27,45,46,47]. Samples were extracted and processed immediately after mouse death; survivor mice were sacrificed (sodium thiopental/ip) after 24 h of treatment. Aseptic thoracotomies were carried out, and through cardiac punctures, blood samples were obtained for quantitative (log_10_ CFU/mL) and qualitative blood cultures. Results were expressed as positive (≥1 CFU present in the plate) or negative. Spleens were aseptically extracted, weighed, and homogenized in sterile saline (Stomacher 80; Tekmar Co., Cincinnati, OH, USA) before quantitative cultures (log_10_ CFU/g) in Columbia agar with 5% sheep blood plates.

### 5.3. Statistical Analysis

Mortality and positive blood cultures were expressed as percentages. Bacterial spleen concentrations (log_10_ CFU/g) and bacterial blood concentrations (log_10_ CFU/mL) were expressed as means ± SD. Differences in bacterial concentrations were compared by analysis of variance and the Dunnet and Tukey’s post hoc tests. Mortality and blood sterility rates between groups were compared by the two-tailed Fisher’s test. A *p*-value < 0.05 was considered significant. SPSS v22.0 was used (SPSS Inc., Chicago, IL, USA).

## Figures and Tables

**Figure 1 antibiotics-11-01212-f001:**
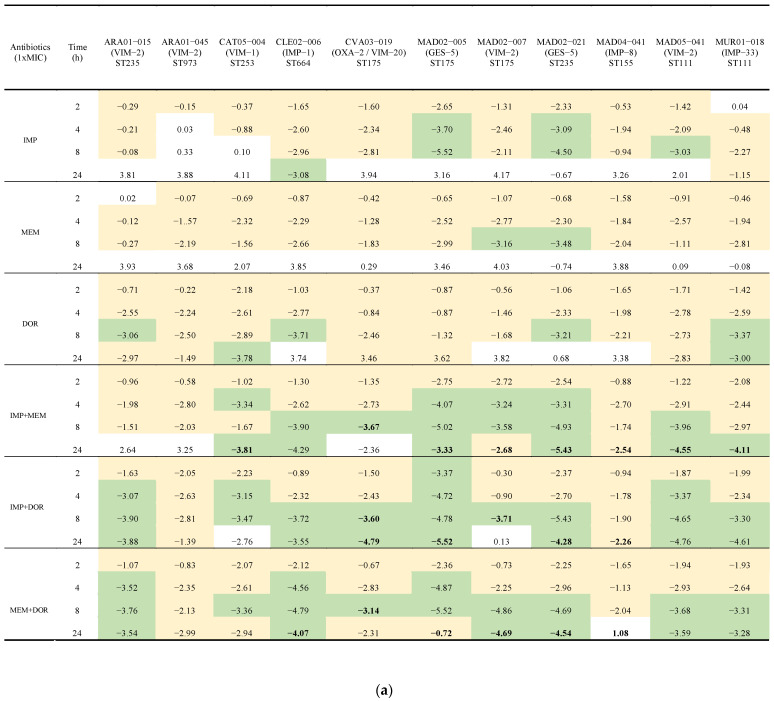
(**a**) Time-kill assays for imipenem, meropenem, and doripenem alone and in combination at MIC concentrations against eleven isolates of carbapenemase-producing *P. aeruginosa* isolates. Results are represented as differences (log_10_ CFU/mL) relative to the initial time point (0 h). Green indicates a >3 log_10_ CFU/mL decrease, yellow a <3 log_10_ CFU/mL decrease. Bold represents synergistic activity with respect to the most active drug alone at that time point. IMP: imipenem; MEM: meropenem; DOR: doripenem. (**b**) Time-kill assays for imipenem, meropenem, and doripenem alone and in combination at MIC concentrations against the four carbapenemase-producing *P. aeruginosa* isolates selected for in vivo studies. Solid circle: growth control; Square: imipenem (IMP); Triangle: meropenem (MEM); Inverted triangle: doripenem (DOR); Empty circle: IMP + MEM; Empty square: IMP + DOR; Empty triangle: MEM + DOR; Dotted line: bactericidal activity.

**Figure 2 antibiotics-11-01212-f002:**
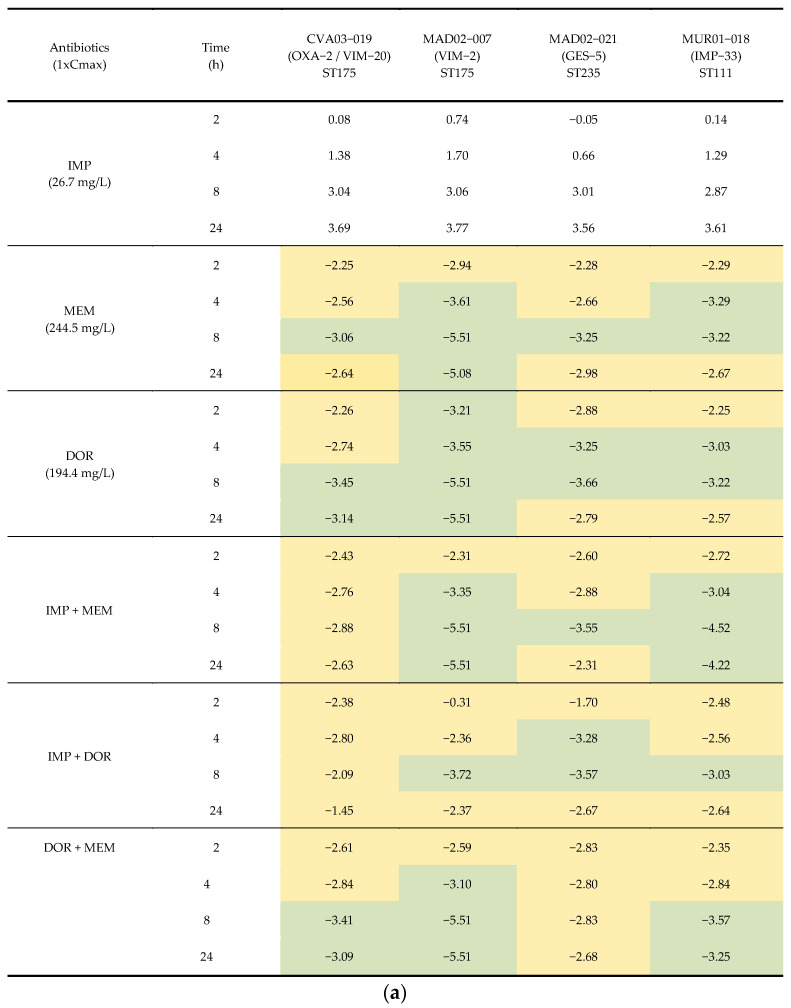
(**a**) Time-kill assays for imipenem, meropenem, and doripenem alone and in combination at maximum mice plasma concentration (Cmax) against the four isolates of carbapenemase-producing *P. aeruginosa* selected for in vivo studies. Results are represented as differences (log_10_ CFU/mL) relative to the initial time point (0 h). Green indicates a >3 log_10_ CFU/mL decrease, yellow a < 3 log_10_ CFU/mL decrease. (**b**) Time-kill assays for imipenem, meropenem, and doripenem alone and in combination at maximum mice plasma concentration (Cmax) against the four of carbapenemase-producing *P. aeruginosa* isolates selected for in vivo studies. Solid circle: growth control; Square: imipenem (IMP); Triangle: meropenem (MEM); Inverted triangle: doripenem (DOR); Empty circle: IMP + MEM; Empty square: IMP + DOR; Empty triangle: MEM + DOR; Dotted line: bactericidal activity.

**Table 1 antibiotics-11-01212-t001:** MICs of imipenem, meropenem, and doripenem and fractional inhibitory concentration indexes (FICi) of dual carbapenem combinations for the eleven carbapenemase-producing *Pseudomonas aeruginosa* clinical strains.

*P. aeruginosa*	ST	Carbapenemases	MIC (mg/L)			ΣFICi (mg/L)		
			IMP	MEM	DOR	IMP + MEM	IMP + DOR	MEM + DOR
ARA01-015	235	VIM-2	128	16	64	1.06	0.56	1.00
ARA01-045	973	VIM-2	64	32	32	0.75	1.06	0.63
CAT05-004	253	VIM-1	128	128	128	1.00	**0.50**	**0.50**
CLE02-006	664	IMP-1	128	64	64	0.75	1.00	0.75
CVA03-019	175	OXA-2/VIM-20	128	32	32	0.75	0.75	0.75
MAD02-005	175	GES-5	64	128	64	0.75	0.75	0.63
MAD02-007	175	VIM-2	128	32	32	**0.50**	1.25	1.00
MAD02-021	235	GES-5	64	128	64	0.63	0.75	0.63
MAD04-041	155	IMP-8	128	16	16	0.75	1.50	0.63
MAD05-041	111	VIM-2	128	16	16	1.00	0.75	**0.50**
MUR01-018	111	IMP-33	256	128	128	1.06	1	**0.25**

ST: Sequence Type. According to EUCAST guidelines: IMP (imipenem) R > 4 mg/L; MEM (meropenem) R > 8 mg/L; DOR (doripenem) R > 4 mg/L. Data on STs and carbapenemases were obtained from a previous study, which includes the complete description of the resistome of the studied strains [23]. Synergistic activity is highlighted in bold. FICi ≤ 0.5 indicates synergy; FICi = 0.5–4 indicates no interaction; and FICi > 4 indicates antagonism.

**Table 2 antibiotics-11-01212-t002:** Bacterial concentrations in spleen and blood and percentages of bacteremia in infected, untreated mice at initiation of treatment and at 24 h after infection.

Hours after Infection	Isolates	Carbapenemases	ST	Spleen (log_10_ CFU/g)	Blood (log_10_ CFU/mL)	Bacteremia (%)	Mortality (%)
2	CVA03-019	OXA-2/VIM-20	175	5.62 ± 0.51	3.35 ± 0.49	100	-
	MAD02-007	VIM-2	175	7.38 ± 0.12	5.69 ± 0.55	100	-
	MAD02-021	GES-5	235	5.64 ± 0.49	3.50 ± 0.28	100	-
	MUR01-018	IMP-33	111	5.90 ± 0.79	2.67 ± 0.46	100	-
24	CVA03-019	OXA-2/VIM-20	175	8.7 ± 0.1	7.2 ± 1.1	100	83
	MAD02-007	VIM-2	175	8.6 ± 1.1	7.1 ± 1.1	100	83
	MAD02-021	GES-5	235	8.3 ± 0.5	7.4 ± 0.6	100	100
	MUR01-018	IMP-33	111	6.5 ± 0.4 ^a,b,c^	2.8 ± 1.0 ^a,b,c^	100	17 ^c^

-: Animals were sacrificed two hours after infection; ^a^: *p* < 0.05 with respect to CVA03-019 isolate; ^b^: *p* < 0.05 with respect to MAD02-007 isolate and ^c^: *p* < 0.05 with respect to MAD02-021 isolate.

**Table 3 antibiotics-11-01212-t003:** In vivo efficacy of imipenem, meropenem, and doripenem, in monotherapy and in combinations, for the experimental peritoneal sepsis model.

Isolates; Clone (Carbapenemase)	Therapy	n	Doses (mg/kg)	Spleen (log_10_ CFU/g)	Blood (log_10_ CFU/mL)	Mortality (%)
**IMP plus MEM**						
CVA03-019; ST175 (OXA-2/VIM-20)	Control	6	-	8.7 ± 0.1	7.2 ± 1.1	83
	IMP	6	30	7.6 ± 1.2	4.9 ± 1.4	17
	MEM	6	300	4.0 ± 0.9 ^a,b^	1.5 ± 1.6 ^a^	17
	IMP + MEM	5		4.4 ± 0.5 ^a,b^	1.6 ± 1.1 ^a,b^	0 ^a^
MAD02-007; ST175 (VIM-2)	Control	6	-	8.6 ± 1.1	7.1 ± 1.1	83
	IMP	6	30	8.5 ± 0.4	6.2 ± 0.8	50
	MEM	6	300	6.6 ± 0.6 ^a, b^	3.4 ± 0.6 ^a,b^	33
	IMP + MEM	6		6.3 ± 0.9 ^a,b^	3.1 ± 0.8 ^a,b^	50
MAD02-021; ST235 (GES-5)	Control	6	-	8.3 ± 0.5	7.4 ± 0.6	100
	IMP	6	30	6.9 ± 1.4	4.1 ± 1.0 ^a^	17 ^a^
	MEM	6	300	3.4 ± 1.1 ^a,b^	1.5 ± 1.3 ^a,b^	17 ^a^
	IMP + MEM	5		0.5 ± 1.1 ^a,b,c^	0.0 ± 0.0 ^a,b^	0 ^a^
MUR01-018; ST111 (IMP-33)	Control	6	-	6.5 ± 0.4	2.8 ± 1.0	17
	IMP	6	30	3.6 ± 0.5 ^a^	0.7 ± 0.9 ^a^	0
	MEM	6	300	3.4 ± 0.6 ^a^	1.5 ± 1.4	0
	IMP + MEM	5		3.6 ± 0.3 ^a^	0.7 ± 1.2	0
**IMP plus DOR**						
CVA03-019; ST175 (OXA-2/VIM-20)	Control	6	-	8.7 ± 0.1	7.2 ± 1.1	83
	IMP	6	30	7.6 ± 1.2	4.9 ± 1.4	17
	DOR	5	150	7.3 ± 1.5	4.8 ± 2.2	40
	IMP + DOR	5		5.9 ± 1.4 ^a^	2.2 ± 0.5 ^a^	0 ^a^
MAD02-007; ST175 (VIM-2)	Control	6	-	8.6 ± 1.1	7.1 ± 1.1	83
	IMP	6	30	8.5 ± 0.4	6.2 ± 0.8	50
	DOR	5	150	7.8 ± 0.6	5.6 ± 0.7	20
	IMP + DOR	5		6.2 ± 1.3	3.0 ± 2.2 ^a^	20
MAD02-021; ST235 (GES-5)	Control	6	-	8.3 ± 0.5	7.4 ± 0.6	100
	IMP	6	30	6.9 ± 1.4	4.1 ± 1.0 ^a^	17 ^a^
	DOR	5	150	4.5 ± 1.7 ^a^	3.0 ± 2.7	0 ^a^
	IMP + DOR	5		2.3 ± 1.3 ^a,b^	2.4 ± 1.8 ^a^	0 ^a^
MUR01-018; ST111 (IMP-33)	Control	6	-	6.5 ± 0.4	2.8 ± 1.0	17
	IMP	6	30	3.6 ± 0.5 ^a,d^	0.7 ± 0.9 ^a^	0
	DOR	6	150	4.8 ± 0.6 ^a^	2.2 ± 1.9	10
	IMP + DOR	5		3.0 ± 1.7 ^a^	0.0 ± 0.0 ^a^	0
**MEM plus DOR**						
CVA03-019; ST175 (OXA-2/VIM-20)	Control	6	-	8.7 ± 0.1	7.2 ± 1.1	83
	MEM	6	300	4.0 ± 0.9 ^a,d^	1.5 ± 1.6 ^a^	17
	DOR	5	150	7.3 ± 1.5	4.8 ± 2.2	40
	MEM + DOR	5		4.8 ± 1.0 ^a^	2.0 ± 1.6 ^a^	20
MAD02-007; ST175 (VIM-2)	Control	6	-	8.6 ± 1.1	7.1 ± 1.1	83
	MEM	6	300	6.6 ± 0.6 ^a^	3.4 ± 0.6 ^a,d^	33
	DOR	5	150	7.8 ± 0.6	5.6 ± 0.7	20
	MEM + DOR	5		6.8 ± 1.0	4.0 ± 1.2 ^a^	20
MAD02-021; ST235 (GES-5)	Control	6	-	8.3 ± 0.5	7.4 ± 0.6	100
	MEM	6	300	3.4 ± 1.1 ^a^	1.5 ± 1.3 ^a^	17 ^a^
	DOR	5	150	4.5 ± 1.7 ^a^	3.0 ± 2.7	0 ^a^
	MEM + DOR	5		2.4 ± 1.4 ^a^	0.0 ± 0.0 ^a^	0 ^a^
MUR01-018; ST111 (IMP-33)	Control	6	-	6.5 ± 0.4	2.8 ± 1.0	17
	MEM	6	300	3.4 ± 0.6 ^a,d^	1.5 ± 1.4	0
	DOR	6	150	4.8 ± 0.6 ^a^	2.2 ± 1.9	10
	MEM + DOR	5		4.0 ± 0.5 ^a^	0.5 ± 0.9	0

IMP: imipenem; MEM: meropenem; DOR: doripenem; -: without treatment; CFU: colony-forming unit; ^a^: *p* < 0.05 with respect to control group; ^b^: *p* < 0.05 with respect to IMP group; ^c^: *p* < 0.05 with respect to MEM group; and ^d^: *p* < 0.05 with respect to DOR group.

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
