# Peer review of "Carbapenem Combinations for Infections Caused by Carbapenemase-Producing Pseudomonas aeruginosa: Experimental In Vitro and In Vivo Analysis"

_antibiotics, 2022, doi:10.3390/antibiotics11091212_

Round 1

Reviewer 1 Report

The authors have evaluated the activity of 3 bactericides alone and in combination to assess their synergistic ability against representative strains of P. aeruginosa.  The authors have done a great job, for which I congratulate them. Below you will find some comments that I would like you to take into account. 

-Line 17: put the full name of P. aeruginosa.

-Line 17-18: mention the selection of 4 reference isolates. Explain why you have decided to select those and not others.

-Line 19: it is already explained in lines 16 and 17 that antibiotics are used alone and in combination to observe their synergistic effect. As a suggestion, it would be convenient to rewrite lines 17-19 to avoid repetition.

-Line 20: put the meaning of the acronym MIC

-Lines 32 and 33 (Keywords): arrange them alphabetically.

-Line 36: remove the graphical abstract and include it in the corresponding section.

-Line 56: detail the meaning of MDR

-Line 59: detail the meaning of VIM, IMP and NDM

-Lines 93-95 (Table 1): put P. aeruginosa, ST and Carbapenemases in boldface type

-Lines 127 to 131 (Figure 2): the figure is of poor quality, perhaps because it is a clipping of the original. Please redo it again to be of the same quality as Figure 1.

-Line 142: p-value in italics.

-Line 147: change cfu to CFU and revise throughout the text (e.g., lines 154, 155, 157,...)

-Line 281: explain which concentrations of the different drugs were used

-Line 289: explain in detail the method 4.1.3.

-Lines 298-299: why higher concentrations were studied for imipenem with respect to the other two antibiotics.

-Line 302: change cfu/mL to CFU/mL (and revise throughout the text, e.g. in lines 318, 319, 347, 359, 362,...) as they appear in lines 123 and 124.

-Line 322: how many animals were used in the in vivo assays? Please indicate this in the text'.

-References: follow the guidelines set by the journal Antibiotics for the bibliography.

Author Response

Dear Reviewer 1,

We are submitting a point-by-point responses to the comments/suggestions raised." We are submitting a word file indicating the changes from the previous submission as file type "antibiotics-1895573 - Revised highlighted in yellow". Also, the new version of the manuscript “antibiotics-1895573 - Revised” without marks is submitted. Finally, we are submitting a new file “Supplementary Figure 1”

Thank you for the consideration of the manuscript.

Sincerely,

María Eugenia

Comments and Suggestions for Authors

The authors have evaluated the activity of 3 bactericides alone and in combination to assess their synergistic ability against representative strains of P. aeruginosa.  The authors have done a great job, for which I congratulate them. Below you will find some comments that I would like you to take into account.

-Line 17: put the full name of P. aeruginosa.

We have modified it as recommended.

-Line 17-18: mention the selection of 4 reference isolates. Explain why you have decided to select those and not others.

Following the Reviewer suggestion we have included it as suggested “According to the widespread global distribution of high-risk clones and carbapenemases, four representative isolates were selected…”

-Line 19: it is already explained in lines 16 and 17 that antibiotics are used alone and in combination to observe their synergistic effect. As a suggestion, it would be convenient to rewrite lines 17-19 to avoid repetition.

Following the Reviewer suggestion we have rewrite it.

-Line 20: put the meaning of the acronym MIC

Following the Reviewer suggestion we have include it.

-Lines 32 and 33 (Keywords): arrange them alphabetically.

Following the Reviewer suggestion we have modified it.

-Line 36: remove the graphical abstract and include it in the corresponding section.

Following the Reviewer comment we have modified its position in the revised manuscript.

A graphical abstract (GA) is an image that appears alongside the text abstract in the Table of Contents.

-Line 56: detail the meaning of MDR

We have included it as recommended by the Reviewer.

-Line 59: detail the meaning of VIM, IMP and NDM

Following the Reviewer request we have included it.

-Lines 93-95 (Table 1): put P. aeruginosa, ST and Carbapenemases in boldface type

We have done it as requested by the Reviewer.

-Lines 127 to 131 (Figure 2): the figure is of poor quality, perhaps because it is a clipping of the original. Please redo it again to be of the same quality as Figure 1.

We have modified it to increase its quality as requested by the Reviewer.

-Line 142: p-value in italics.

We have modified it as requested.

-Line 147: change cfu to CFU and revise throughout the text (e.g., lines 154, 155, 157,...)

As requested by the Reviewer we have modified it throughout the entire manuscript.

-Line 281: explain which concentrations of the different drugs were used

We have included the concentrations in the method sections 4.1.3 following the suggestion bellow.

-Line 289: explain in detail the method 4.1.3.

Following the reviewer request we have detailed it.

-Lines 298-299: why higher concentrations were studied for imipenem with respect to the other two antibiotics.

The reason is that for one of the strains MUR01-018, ST111 (IMP-33) tested the MIC concentration for imipenem was 256 mg/L and we like always to start this assays with a dilution above the MIC to ensure that this has not change and that the dilution made is correct. In the case of the other antibiotics (meropenem and doripenem the highest MIC obtained for the studied strains was 128 mg/L.

-Line 302: change cfu/mL to CFU/mL (and revise throughout the text, e.g. in lines 318, 319, 347, 359, 362,...) as they appear in lines 123 and 124.

As requested by the Reviewer we have modified it throughout the entire manuscript.

-Line 322: how many animals were used in the in vivo assays? Please indicate this in the text'.

We have included it in the text as requested in the 4.2.1. Animals Section.

-References: follow the guidelines set by the journal Antibiotics for the bibliography.

We have revised and modified it following the author guidelines as suggested by the Reviewer.

Reviewer 2 Report

Great article, minor edits needed

1-    Title: suggestion “Efficacy of carbapenem combinations for infections caused by carbapenemase-producing Pseudomonas aeruginosa: Experimental in-vivo and in-vitro study” please make sure that I classified the study correctly as in vivo and in vitro 

2-    Introduction, line 44, “morbimortality” I recommend changing this word with something else because it is not clear

3-     The paragraph starting with "A recent study in India showed that m….” combine it with the previous paragraph since it is related to antimicrobial resistance, 

4-    Method, 4.1.1 Bacterial Isolates’ _Characterization starting from here “A 2021 multicenter study…. Until the end of the paragraph” this appears to be a discussion. If you used a methodology of the study you cited, just write “ we chose x and add the reference without discussing the literature here. So please reformat this section 

5-    In the method section, If there are details that can be reduced it would be great

6-    Discussion: you need to add a paragraph “typically last paragraph” for the limitations of your research

7-    In the discussion, can elaborate on the safety profile for using combination carbapenem? Clinically carbapenem increases superinfection, so are there any safety data? Perhaps add it to the limitations  

8-    Journal formatting: I think the conclusion should be a separate section and not part of the discussion 

9-     Abstract should be structured background, method, results, conclusion 

Author Response

Dear Reviewer 2,

We are submitting a point-by-point responses to the comments/suggestions raised." We are submitting a word file indicating the changes from the previous submission as file type "antibiotics-1895573 - Revised highlighted in yellow". Also, the new version of the manuscript “antibiotics-1895573 - Revised” without marks is submitted. Finally, we are submitting a new file “Supplementary Figure 1”

Thank you for the consideration of the manuscript.

Sincerely,

María Eugenia

Comments and Suggestions for Authors

Great article, minor edits needed

1-    Title: suggestion “Efficacy of carbapenem combinations for infections caused by carbapenemase-producing Pseudomonas aeruginosa: Experimental in-vivo and in-vitro study” please make sure that I classified the study correctly as in vivo and in vitro

Following the Reviewer suggestion, as well as another comment of the Reviewer 4, we have modified the title in the revised manuscript.

2-    Introduction, line 44, “morbimortality” I recommend changing this word with something else because it is not clear

Following the Reviewer suggestion we have modified it to make it clearer.

3-     The paragraph starting with "A recent study in India showed that m….” combine it with the previous paragraph since it is related to antimicrobial resistance,

Following the Reviewer suggestion we have combined them.

4-    Method, 4.1.1 Bacterial Isolates’ _Characterization starting from here “A 2021 multicenter study…. Until the end of the paragraph” this appears to be a discussion. If you used a methodology of the study you cited, just write “we chose x and add the reference without discussing the literature here. So please reformat this section

Following the Reviewer suggestion we have changed the section 5.1.1 Bacterial Isolates’ _Characterization in the revised manuscript. This change includes moving some sentences and references to the last paragraph of the Discussion Section.

5-    In the method section, If there are details that can be reduced it would be great

About this Reviewer's suggestion, we find ourselves in the situation that the Reviewer 1 asks us to further develop some of the points of the Methods Section, so we have tried to do what both suggest, reducing some sub-section of this section, as in the previous comment of the reviewer, and developing some of those that we have being asked to do by the Reviewer 1.

6-    Discussion: you need to add a paragraph “typically last paragraph” for the limitations of your research

Following the Reviewer suggestion, we have included a limitation paragraph at the end of the Discussion Section.

7-    In the discussion, can elaborate on the safety profile for using combination carbapenem? Clinically carbapenem increases superinfection, so are there any safety data? Perhaps add it to the limitations 

Following the Reviewer comment we have included a paragraph about the safety of the use of double carbapenem therapy in a new paragraph of the Discussion Section.

8-    Journal formatting: I think the conclusion should be a separate section and not part of the discussion

Following the Reviewer comment we have separate the conclusion into a new section.

9-     Abstract should be structured background, method, results, conclusion

We have re-checked the author instructions of the Antibiotics journal, to accomplish the Reviewer suggestion and have found that they specify that abstract should be a single paragraph and should follow the structured background, method, results, and conclusion, but without headings.

Reviewer 3 Report

well-designed scientific work of clinical interest. Figure 2 is low resolution.

Author Response

Dear Reviewer 3,

We are submitting a point-by-point responses to the comments/suggestions raised." We are submitting a word file indicating the changes from the previous submission as file type "antibiotics-1895573 - Revised highlighted in yellow". Also, the new version of the manuscript “antibiotics-1895573 - Revised” without marks is submitted. Finally, we are submitting a new file “Supplementary Figure 1”

Thank you for the consideration of the manuscript.

Sincerely,

María Eugenia

Comments and Suggestions for Authors

Well-designed scientific work of clinical interest. Figure 2 is low resolution.

Following the Reviewer suggestion we have improved the Figure 2.

Reviewer 4 Report

Dear Authors

Greetings

Please follow the suggestions on the attached doc *reading it with Adobe.

Kind regards

Author Response

Comments and Suggestions for Authors

Dear Authors

Greetings

Please follow the suggestions on the attached doc *reading it with Adobe.

Kind regards

Dear Reviewer 4,

Thanks for all your suggestions. We have copy in this document them from the attached document to respond to them one by one. We are submitting a word file indicating the changes from the previous submission as file type "antibiotics-1895573 - Revised highlighted in yellow". Also, the new version of the manuscript “antibiotics-1895573 - Revised” without marks is submitted. Finally, we are submitting a new file “Supplementary Figure 1”

Thank you for the consideration of the manuscript.

Sincerely,

María Eugenia

Efficacy or effectiveness???

Efficacy can be demonstrated in an explanatory, ie, a randomized controlled trial (RCT) completed under ideal study conditions. Effectiveness can be demonstrated in an observational, ie, a pragmatic controlled trial (PCT) completed under real-world conditions. It is impossible to design a trial which can detect efficacy and effectiveness simultaneously.

Porzsolt F, Rocha NG, Toledo-Arruda A, Thomaz T, Moraes C, Bessa-Guerra T, Leão M, Migowski A, da Silva ARA, Weiss C. Efficacy and effectiveness trials have different goals, use different tools, and generate different messages. Pragmat Obs Res. 2015;6:47-54 https://doi.org/10.2147/POR.S89946

Following the Reviewer comment and combining it with other suggestion of the Reviewer 2 we have modified the tittle.

According this the tittle of the article must be changed!!!

As specified in the previous comment, we have changed the title in the revised manuscript.

Please consider this point after read the following works:

Song, X., Wu, Y., Cao, L., Yao, D., & Long, M. (2019). Is Meropenem as a Monotherapy Truly Incompetent for Meropenem-Nonsusceptible Bacterial Strains? A Pharmacokinetic/Pharmacodynamic Modeling With Monte Carlo Simulation. Frontiers in microbiology, 10, 2777. https://doi.org/10.3389/fmicb.2019.02777

Cathrine McKenzie, Antibiotic dosing in critical illness, Journal of Antimicrobial Chemotherapy, Volume 66, Issue suppl_2, April 2011, Pages ii25–ii31, https://doi.org/10.1093/jac/dkq516

Following the Reviewer suggestion, we have included a comment from Song et al. PK/PD modeling study, in the first paragraph of the Discussion Section, which supports the results obtained with meropenem in monotherapy in our in vivo experiments.  

Image has low quality

Following the Reviewer suggestion, we have improved the graphical abstract quality.

It is identified as a common coinfecting pathogen in COVID-19 patients causing exacerbation of illness. In our hospital, P. aeruginosa is one of the top coinfecting bacteria identified among COVID-19 patients. Qu J, Cai Z, Liu Y, Duan X, Han S, Liu J, Zhu Y, Jiang Z, Zhang Y, Zhuo C, Liu Y, Liu Y, Liu L and Yang L (2021) Persistent Bacterial Coinfection of a COVID-19 Patient Caused by a Genetically Adapted Pseudomonas aeruginosa Chronic Colonizer. Front. Cell. Infect. Microbiol. 11:641920. doi:

Following the reviewer suggestion, we have included this reference in the first paragraph of the Introduction Section.

This is the only time the authors speak of resistance at work...

We do not quite understand what the Reviewer means with the sentence “This is the only time the authors speak of resistance at work…”, at the beginning of the second paragraph of the Introduction. After an initial paragraph in the Introduction Section on the clinical importance of P. aeruginosa infections, with the following paragraphs we aimed to detail the global importance of carbapenem-resistant and carbapenemase-producing strains, since we focused in them to analyze the effectiveness of dual carbapenem treatment, both in vitro and in a mouse model of infection.

IT IS DISCUSSION!

Following the Reviewer suggestion, we have modified this paragraph. We have shortened it, just using the references to introduce the reason of the present study, and we have also moved this information and the references to the Discussion Section.  

Results is not italicized

Following the Reviewer comment we have corrected it.

The authors can explory it more. I suggest to use time-kill curves for single-drug and two-drug combinations. Each graph shows the time-kill curve for a bactericidal-bacteriostatic drug pair/synergism and the constituent individual drugs.

Following the Reviewer suggestion, we have included a Figure X in the revised manuscript with time-kill curves for single-drug and two-drug combinations of the four selected strains used in the animal model at MIC and Cmax concentrations. Moreover, we are submitting a material Supplementary Figure X with the time-kill curves for single-drug and two-drug combinations in the other seven strains used in the in vitro studies, at MIC concentrations.

Image has low quality

Following the Reviewer suggestion, we have improved the Figure 2 quality.

Results is not italicized

Following the Reviewer comment we have corrected it.

Also they had use female??

Do you think it can induce the results?

RESEARCH ARTICLE – PHARMACOKINETICS, PHARMACODYNAMICS AND DRUG TRANSPORT AND METABOLISM| VOLUME 104, ISSUE 5, P1848-1855, MAY 01, 2015

PEGylated Interferon Displays Differences in Plasma Clearance and Bioavailability Between Male and Female Mice and Between Female Immunocompetent C57Bl/6J and Athymic Nude Mice. Cornelia B. Landersdorfer, Suzanne M. Caliph, David M. Shackleford, David B. Ascher, Lisa M. Kaminskas, DOI:https://doi.org/10.1002/jps.24412

Yes, they also used female C57BL/6J in the murine sepsis model by carbapenemase-producing A. baumannii [16].

We certainly believe that the animal gender impact in the results, as it was published, among others, in “Biological sex influences susceptibility to Acinetobacter baumannii pneumonia in mice. JCI Insight. 2020. doi: 10.1172/jci.insight.132223)”. There are others aspects that could affect the results when planning animal’s models such as the age (doi: 10.18632/aging.101495.) or mixing animals from different laboratories, even of the same strain, which has also been proven to affect the research results.

In order to include it in the revised manuscript, we have added a limitation paragraph in the Discussion Section.

In your article conclusion, go back over the main points (often the information you included in subheadings) and reiterate them. You're summarizing the important information so your readers know what to take away from the piece. We already talked about restating your thesis, but you also want to cover your sub-points.

In the Conclusions we have addressed the in vitro and in vivo results, which are the sub-headings of the Results Section, discussed in the corresponding section.

Moreover, we are not sure that this comment refers to the manuscript, because of the comment of the Reviewer includes the sentence “We already talked about restating your thesis, …”,

Results is not italicized

Following the Reviewer comment we have corrected it.

Results is not italicized

Following the Reviewer comment we have corrected it.

RESEARCH ARTICLE – PHARMACOKINETICS, PHARMACODYNAMICS AND DRUG TRANSPORT AND METABOLISM| VOLUME 104, ISSUE 5, P1848-1855, MAY 01, 2015

PEGylated Interferon Displays Differences in Plasma Clearance and Bioavailability Between Male and Female Mice and Between Female Immunocompetent C57Bl/6J and Athymic Nude Mice

Cornelia B. Landersdorfer

Suzanne M. Caliph

David M. Shackleford

David B. Ascher

Lisa M. Kaminskas

DOI:https://doi.org/10.1002/jps.24412

We believe that this Reviewer's comment follows the previous one in which he/she asked if we believed that gender could influence the results. As we responded before, it does, as do other biological and environmental factors.

Preclinical models of infection have limitations such as passing strict ethical committees for the use of experimental animals, which limits, among other aspects, the ability to perform these studies in more animal groups, since we must comply with the 3R rules (Hubrecht & Carter, 2019), that prevents us from increasing the numbers of animals to test the same hypothesis in male mice.

At first, the references made no attempt at standardization defined according to the journal's rules

Following the Reviewer suggestion, we have corrected the mistakes in the reference list to accomplish with the journal rules.

I have to suggest to include the following works into the references:

  • Fred C. Tenover, David P. Nicolau & Christian M. Gill (2022) Carbapenemase-producing Pseudomonas aeruginosa –an emerging challenge, Emerging Microbes & Infections, 11:1, 811-814, DOI: 10.1080/22221751.2022.2048972
  • Preeti Pachori, Ragini Gothalwal, Puneet Gandhi,Emergence of antibiotic resistance Pseudomonas aeruginosa in intensive care unit; a critical review, Genes & Diseases, Volume 6, Issue 2, 2019, Pages 109-119, ISSN 2352-3042, https://doi.org/10.1016/j.gendis.2019.04.001.(https://www.sciencedirect.com/science/article/pii/S2352304219300170)
  • Qu J, Cai Z, Liu Y, Duan X, Han S, Liu J, Zhu Y, Jiang Z, Zhang Y, Zhuo C, Liu Y, Liu Y, Liu L and Yang L (2021) Persistent Bacterial Coinfection of a COVID-19 Patient Caused by a Genetically Adapted Pseudomonas aeruginosa Chronic Colonizer. Front. Cell. Infect. Microbiol. 11:641920. doi: 10.3389/fcimb.2021.641920
  • Buehrle, D. J., Shields, R. K., Clarke, L. G., Potoski, B. A., Clancy, C. J., & Nguyen, M. H. (2016). Carbapenem-Resistant Pseudomonas aeruginosa Bacteremia: Risk Factors for Mortality and Microbiologic Treatment Failure. Antimicrobial agents and chemotherapy, 61(1), e01243-16. https://doi.org/10.1128/AAC.01243-16

Following the Reviewer suggestion, we have included three of these references, two in the first paragraph of the Introduction Section and one in the last paragraph of the Discussion Section.